# Microbiome of *Hyalomma dromedarii* (Ixodida: Ixodidae) Ticks: Variation in Community Structure with Regard to Sex and Host Habitat

**DOI:** 10.3390/insects16010011

**Published:** 2024-12-27

**Authors:** Nighat Perveen, Alejandro Cabezas-Cruz, Daniil Iliashevich, Lianet Abuin-Denis, Olivier Andre Sparagano, Arve Lee Willingham

**Affiliations:** 1Department of Biology, College of Science, United Arab Emirates University, Al-Ain P.O. Box 15551, United Arab Emirates; nighat.p@uaeu.ac.ae; 2Department of Veterinary Medicine, College of Agriculture and Veterinary Medicine, United Arab Emirates University, Al-Ain P.O. Box 15551, United Arab Emirates; daniil_l@uaeu.ac.ae; 3ANSES, INRAE, Ecole Nationale Vétérinaire d’Alfort, UMR BIPAR, Laboratoire de Santé Animale, F-94700 Maisons-Alfort, France; alejandro.cabezas@vet-alfort.fr (A.C.-C.); labuind@gmail.com (L.A.-D.); 4Animal Biotechnology Department, Center for Genetic Engineering and Biotechnology, Avenue 31 Between 158 and 190, Havana CU-10600, Cuba; 5Department of Infectious Diseases and Public Health, Jockey Club College of Veterinary Medicine and Life Sciences, City University of Hong Kong, Kowloon, Hong Kong SAR, China; olivier.sparagano@cityu.edu.hk; 6UK Management College, Manchester M11 1AA, UK

**Keywords:** camel ticks, *Hyalomma dromedarii*, vector, microbiome, bacterial communities, habitat

## Abstract

Camels are highly valued animals in the United Arab Emirates (UAE). The camel tick, *Hyalomma dromedarii*, may transmit various pathogens to animals and humans, leading to economic losses to the livestock industry. We analyzed the bacterial communities of male and female *H. dromedarii* ticks collected from different habitats to investigate how sex and host habitat influence the tick microbiome. Our findings revealed that the microbiomes of *H. dromedarii* ticks vary by sex and habitat and contain pathogenic bacteria along with endosymbionts. Understanding the microbial ecology of *H. dromedarii* is essential for preventing the spread of tick-borne pathogens across habitats and protecting both humans and animals in the region.

## 1. Introduction

Vector-borne infectious diseases impose a significant burden on animal and human health. In the past four decades, the emergence and re-emergence of many vector-borne pathogens has created new challenges for public health [1,2,3]. Microbiota in arthropod vectors may shape vector competence by acquiring and transmitting pathogens to hosts [4,5,6,7,8]. The vector–pathogen relationship has been disturbed, mostly due to changes in climate, land use, animal host communities, human living conditions, and societal factors that result in the expansion of the vectors’ and pathogens’ distributions [1]. The emergence and re-emergence of tick-borne diseases with a change in their epidemiology, including prevalence, pathogenicity, and geographic distribution, is a critical area of query that needs to be addressed immediately. Parasitic and microbial ecology is an emerging discipline because of the potential role of parasites in the regulation of host populations and their impact on the balance and functioning of ecosystems [9]. Ticks are arthropod vectors of many pathogens, including viruses, bacteria, and protozoa, and facilitate transmission of infections among host species [10,11]. Farming and other artificial animal settings may not only impact tick prevalence and abundance on camels and other livestock but also impact the microbial communities’ composition inside tick vectors. Ticks may harbor a diverse range of commensal, endosymbiotic [12], and pathogenic microorganisms [13]. Commensals and symbionts microorganisms may have several detrimental, neutral, or beneficial effects on their tick hosts [14,15]. Moreover, these can play various roles in nutritional adaptation, growth and reproduction, resistance against environmental stress, and immunity [4,15,16]. In the case of bacterial genera, ticks can carry and transmit *Anaplasma*, *Borrelia*, *Coxiella*, *Ehrlichia*, *Francisella*, *Rickettsia,* etc. [17,18,19], and these bacteria are adapted to undergo development in the tick vector for at least a portion of their lifecycle.

*Hyalomma dromedarii*, commonly known as the camel tick, is a significant ectoparasite primarily associated with camels and prevalent in arid and semi-arid regions, particularly in the Middle East, North Africa, and parts of Asia [20,21]. Fast development in the Middle East region has resulted in an associated growth in the farming industry throughout the region to meet an increasing demand for camel milk and meat. *Hyalomma dromedarii* is not only a nuisance to its hosts but also a vector for various pathogens which can cause diseases in animals and humans, for example Crimean–Congo hemorrhagic fever virus (CCHFV) [22]; Dhori virus [23]; tropical theileriosis caused by *T. annulata* and *T. camelensis* [23,24]; Sindbis, Chick Ross, and Kadam viruses [25]; Q fever caused by *Coxiella burnetii* [26]; and spotted fever rickettsia [27,28,29,30,31]. In the Middle East and North Africa (MENA) Region, *H. dromedarii* is reported to have a high prevalence on camels [21,32,33]. Understanding the microbial communities associated with *H. dromedarii* is crucial for comprehending its role in pathogen transmission and its overall impact on host health.

Tick microbiomes may differ between the sexes, life stages, host habitats, and livestock environmental settings due to tick feeding behavior, host parasitism, movement, and reproductive period [34,35,36]. There is emerging evidence that diversity of microbial communities changes due to environmental conditions including temperature, suggesting seasonality of the microbiota, which could in turn be linked with seasonality of pathogen transmission [37]. Traditional methods of studying tick microbiota have been limited in scope, often missing many of the microbial players involved. However, the advent of next-generation sequencing (NGS) technologies has revolutionized this field by providing a comprehensive and detailed analysis of microbial communities [38,39]. NGS methods include 16S rRNA gene sequencing, shotgun metagenomic sequencing, and RNA sequencing that enable researchers to identify and quantify the diverse array of microorganisms present within ticks [40]. Furthermore, NGS can uncover previously undetectable microbes, elucidate complex microbial interactions, and provide insights into the ecological and evolutionary dynamics of microbes [41]. Using the 16S gene sequence method in NGS enables this tick research area and enhances our understanding of the tick’s microbiome to develop targeted strategies for tick control and tick-borne disease management. NGS technology has revolutionized genomic research by reducing the cost for analysis of substantial amounts of genetic data [42]. Microbiome studies mostly utilize the Illumina MiSeq platform, which is reported to be more cost-effective and precise, and the V3-V4 hypervariable region is generally selected for work on the MiSeq because it provides sufficient information for taxonomic profiling of microbial communities with a lower error rate [43], enabling a better assessment of the variety of circulating microbes [41]. *Hyalomma dromedarii* harbors a variety of microbes, including endosymbionts that often form complex interactions with pathogenic microbes. Previously, a *Francisella*-like endosymbiont was reported in *H. dromedarii* ticks [13,44,45]. In another study on evaluating temporal changes in bacterial communities in *H. dromedarii* using high-throughput sequencing, the genus *Francisella* was significantly positively correlated with *Rickettsia* [37]. However, it was significantly negatively correlated with *Acinetobacter*, *Corynebacterium,* and *Escherichia* [46].

So far, there are no studies on the comparison of microbial communities’ diversity in male and female *H. dromedarii* ticks and in different habitats in the UAE. We assumed that bacterial communities differ in male and female *H. dromedarii* ticks feeding on animals in different habitats. Therefore, the aim of this study was (i) to assess the microbiota associated with *H. dromedarii* male and female ticks in the UAE and (ii) to determine patterns of microbial communities in camel ticks collected from different habitats (farms vs. slaughterhouses). This will lead to better understanding of how gender and host habitats impact the localization of various bacterial communities.

## 2. Materials and Methods

### 2.1. Ethics Statement

Permission for tick collection was obtained from the relevant authorities. Tick sampling was conducted in accordance with the experimental protocol approved by the Animal Research Ethics Committee of the UAE University (ethical approval # ERA_2022_1647).

### 2.2. Tick Sampling

In 2022 and 2023, in a cross-sectional study, ticks were collected manually from camels at different farms from seven locations in the UAE (Nahil, Al Foah, Sieh Al Hama, Al Hiyar, Al-Wagan, Al Khazna slaughterhouse, and Al Bawadi Livestock Market). Five camels were selected randomly at each farm. At each location, 10 ticks were collected from each camel. Ticks were placed in Eppendorf tubes (50 mL) and brought to the Parasitology Laboratory at the Department of Veterinary Medicine, College of Agriculture and Veterinary Medicine, UAE University, Al Ain. One male, female, and nymph were selected out of the 10 ticks collected per host for DNA extraction. Consequently, 350 ticks in total were gathered from seven locations. No nymphs were collected from the livestock market. Forty partially engorged males, females, and nymphs were used for this study. The tick samples were stored in a −80 °C freezer until DNA extraction.

### 2.3. Tick Identification, Genomic DNA Extraction, and Pooling of Samples

All ticks were morphologically identified as *H. dromedarii* [47,48]. DNA was extracted individually from 40 ticks (males (15), females (15), and nymphs (10)). As per published protocol, each tick was washed in ethanol (70%) and then for five minutes in deionized water to remove environmental contaminants [49]. Each whole tick was crushed manually using a sterile Kimble Kontes pellet pestle (Thermo Fisher, Waltham, MA, USA) inside a sterile 1.5 mL microcentrifuge tube. A DNeasy Blood & Tissue Kit (Qiagen, Hilden, Germany) was used to extract genomic DNA from each individual tick following the manufacturer’s protocol. DNA quality and concentration were determined with a NanoDrop 2000 spectrophotometer (ThermoFisher Scientific™, Waltham, MA, USA). In addition, DNA quality was assessed using a Qubit 2.0 fluorometer (ThermoFisher Scientific™) and stored in a −20 °C freezer until used. Prior to sequencing, DNA samples were pooled according to sex/stages and habitat, which resulted in eight DNA pools.

### 2.4. Sequencing and Bioinformatics

To determine the microbial community composition in camel ticks, we conducted a 16S ribosomal RNA gene-based analysis. Eight DNA samples (pools) were shipped to Macrogen Inc. (Seoul, South Korea) for NGS. However, four DNA pools (FM, FF, SM, SF) (Table 1) passed the quality control check. The primer set used for amplification of the hypervariable V3-V4 region included Bakt_341F: CCT ACGGGNGGC WGC AG and Bakt_805R: GAC TAC HVGGG TAT CTA ATC C [50]. After conducting PCR using the Herculase II Fusion DNA polymerase Nextera XT Index Kit V2 (Agilent Technologies, Inc., Santa Clara, CA, USA), sequencing was performed on an Illumina MiSeq platform with a read length of 301 bp. Paired-end FASTQ sequence reads were merged using fast length adjustment of short reads (FLASH) version 1.2.11 [51], and CD-HIT-OTU [52] was used to cluster the reads from both sexes of ticks and habitats using the default options (Appendix A). We filtered out low quality reads, trimmed extra-long tails, and identified chimeric reads and then clusters reads into operational taxonomic units (OTUs) with a cutoff of 97% similarity. The taxonomic assignment of OTUs was performed using Quantitative Insights into Microbial Ecology (QIIME 2) through the assign_taxonomy.py. script [53], and the assignment was based on the Basic Local Alignment Search Tool (BLAST) [54] search in the National Center for Biotechnology Information (NCBI) 16S microbial database. Alpha diversity was explored using alpha_diversity.py. The taxonomic abundance count was calculated using Microsoft Excel (Microsoft Corp., Redmond, WA, USA) to estimate abundance ratios at different levels, including phylum, class, order, family, and genus level. The current study sequences were submitted in the NCBI Sequence Read Archive (SRA) under the BioProject ID PRJNA1113868.

### 2.5. Diversity Indices

To test for differences in bacterial diversity between farm and slaughterhouse samples, we conducted analyses of alpha and beta diversity. Alpha diversity refers to the diversity within a single community. We explored the community richness and evenness using observed features [55], Faith’s phylogenetic diversity (Faith_pd) [56], Shanon entropy index [57], and Pielou’s evenness index [58], respectively, as alpha diversity measures. Beta diversity measures the variability of samples between different conditions to assess the similarity of the communities. To measure the microbial beta diversity, the Bray–Curtis dissimilarity index was used [59]. The Bray–Curtis index was used to evaluate dissimilarity between two conditions, considering the relative abundance of taxa. The Bray–Curtis was measured using the ‘vegan’ package [60] implemented in RStudio (https://www.R-project.org/, accessed on 8 October 2024). Cluster analysis was performed with the Jaccard coefficient of similarity using Vegan implemented (https://github.com/vegandevs/vegan, accessed on 8 October 2024) in RStudio (https://www.R-project.org/, accessed on 8 October 2024). The Jaccard distance is represented between 0 and 2, and lines are proportional to this distance.

### 2.6. Inference of Bacterial Co-Occurrence Networks

We constructed co-occurrence networks for farm and slaughterhouse datasets based on taxonomic profiles at the family and genus levels. The networks provide a graphical representation of microbial community assemblies, with nodes representing bacterial taxa and edges denoting correlations between taxa. To determine correlation strength, we employed the Sparse Correlations for Compositional data (SparCC) method [61] in R v.4.3.1 R Core Team, 2023 and performed using the RStudio environment (RStudio Team, 2020). Taxonomic data tables were used to calculate the correlation matrix. Node colors were assigned based on modularity class metric values, and the node size was proportional to the eigenvector centrality of each taxon. The blue colors of the edges represent positive (average weight > 0.75), while red colors indicate negative correlations (average weight < 0.75).

Various network topological features were computed and visualized using Gephi 0.9.5 [62]. These features include the number of nodes and edges, network diameter (the shortest path between the two most separated nodes), modularity (indicating the strength of network division into modules), average degree (the average number of edges per node), weighted degree (the sum of edge weights connected to a node), and clustering coefficient (indicating the tendency of nodes to form clusters) [63].

To investigate the interactivity of *Francisella* within the community, we determined their direct relationships with other bacterial microbiome members. For this purpose, sub-networks were constructed to visualize direct positive and negative associations. The analyses were conducted in Gephi 0.95 [62], with the strength of the edges represented using the SparCC weight.

### 2.7. Statistical Analyses

We determined richness (total number of genera, based on OTUs obtained for each genus), Faith’s phylogenetic diversity (Faith_pd), Pielou’s evenness, and Shannon Wiener Index using the Kruskal–Wallis test (*p* ≤ 0.05) within QIIME 2 [64]. To determine the pattern of bacterial diversity in ticks collected from different host habitats, Principal Coordinates Analysis (PCoA) was conducted and visualized using the PAST 5.27 paleontological statistics software package (Øyvind Hammer, Natural History Museum, University of Oslo, Oslo, Norway, ohammer@nhm.uio.no) [65]. The OTU count of each genus was entered, and the samples were categorized by habitat (farm or slaughterhouse). The Eigenvalues were examined to determine the magnitude of variation [66]. For all tests, the value of α was set at 0.05. The layout of the working procedure for data collection and analysis is illustrated in Figure 1.

## 3. Results

### 3.1. Microbial Community Composition in H. dromedatii Ticks Across Different Host Habitats

We obtained 151,168 read counts (an average 37,792 sequences per sample; minimum 33,946 sequences per sample; and maximum 41,658 sequences per sample) that formed 237 OTUs (clustered at 97% identity) (Appendix A), representing 11 phyla, 22 classes, 77 families, and 164 genera. The phyla Actinomycetota, Bacillota, Bacteroidota, Pseudomonadota, and Fusobacteriota were the most abundant in the taxonomic profiling of the bacteria from *H. dromedarii* sampled from the different habitats. The phylum Pseudomonadota was the most abundant in almost all male and female tick samples collected from farms and slaughterhouses, while Myxococcota had the least abundance (Appendix A). Out of 22 bacterial classes, 17 classes were abundant, including Actinomycetes, Nitriliruptoria, Bacilli, Clostridia, Erysipelotrichia, Tissierellia, Bacteroidia, Flavobacteriia, Balneolia, Cyanophyceae, Deinococci, Fusobacteriia, Alphaproteobacteria, Betaproteobacteria, Gammaproteobacteria, Spartobacteria, and Verrucomicrobiae. Gammaproteobacteria was recorded as the dominant class in male ticks (90.90%) collected from farms and female ticks (72.68%) from the slaughterhouse (Appendix A). The orders Enterobacterales (89.83%) and Mycobacteriales (31.74%) were abundant in ticks from farms, whereas Thiotrichales (70.55%) and Bacillales (25.35%) were abundant in slaughterhouse ticks (Appendix A). Out of 77 families, taxonomic assignment showed that 44 were abundant, namely, Actinomycetaceae, Kytococcaceae, Microbacteriaceae, Micrococcaceae, Corynebacteriaceae, Pseudonocardiaceae, Euzebyaceae, Bacillaceae, Staphylococcaceae, Aerococcaceae, Carnobacteriaceae, Lactobacillaceae, Streptococcaceae, Clostridiaceae, Eubacteriaceae, Lachnospiraceae, Oscillospiraceae, Peptostreptococcaceae, Erysipelotrichaceae, Turicibacteraceae, Peptoniphilaceae, Porphyromonadaceae, Prevotellaceae, Rikenellaceae, Weeksellaceae, Balneolaceae, Nodosilineaceae, Deinococcaceae, Fusobacteriaceae, Methylobacteriaceae, Nitrobacteraceae, Rhizobiaceae, Paracoccaceae, Sphingomonadaceae, Comamonadaceae, Oxalobacteraceae, Neisseriaceae, Enterobacteriaceae, Morganellaceae, Moraxellaceae, Pasteurellaceae, Pseudomonadaceae, Francisellaceae, and Akkermansiaceae (Appendix A). Francisellaceae was highly abundant in ticks collected from slaughterhouses, followed by Corynebacteriaceae in ticks from farms. Staphylococcaceae was detected with high relative abundance in ticks collected from slaughterhouses; however, Morganellaceae and Moraxellaceae were reported with high relative abundance in ticks from farms. In the microbiome of *H. dromedarii* collected from different farms and slaughterhouses, the relative abundance of genera was highly variable among different habitats. *Corynebacterium*, *Staphylococcus*, *Peptoniphilus*, and *Moraxella* were abundant in all habitats. The dominant bacterial genus was *Francisella,* followed by *Staphylococcus,* in slaughterhouse ticks, whereas *Corynebacterium* was recorded with a high relative abundance, followed by *Moraxella,* in farm ticks. However, *Streptococcus*, *Anaerococcus*, *Holdemania*, *Prevotella*, *Epilithonimonas*, *Fusobacterium*, and *Francisella* were abundant in ticks collected from slaughterhouses (Figure 2; Appendix A).

### 3.2. Microbial Community Composition in H. dromedatii Ticks’ Sex

We have investigated for the first time whether the bacterial communities differ with regard to male and female ticks. In the case of bacterial phyla, Pseudomonadota was most abundant (92.02%), followed by Bacillota (47.35%) and Actinomycetota (35.83%). Balneolota, Gemmatimonadota, and Myxococcota were absent in female ticks collected from both farm and slaughterhouse habitats (Appendix A). We identified all bacterial phyla from slaughterhouse male ticks. Gammaproteobacteria was the most dominant class (90.90%), as mentioned above, in male farm ticks, whereas Tissierellia, Bacilli, and Actinomycetes were all low in terms of relative abundance (1.57, 1.94, and 4.23%, respectively). In addition, the classes Bacteroidia, Flavobacteriia, Balneolia, Cyanophyceae, Deinococci, Fusobacteriia, Alphaproteobacteria, Spartobacteria, and Verrucomicrobiae were absent in male ticks collected from farms; however, Verrucomicrobiae was absent only in male ticks collected from slaughterhouses. The families Corynebacteriaceae (31.74%), Moraxellaceae (22.12%), Peptoniphilaceae (14.79%), and Staphylococcaceae (11.05%) were the predominant families in female ticks collected from farms, whereas Morganellaceae (89.83%) was dominant in male farm ticks. *Francisellaceae* had the highest relative abundance in both female and male ticks from slaughterhouses (70.55% and 29.56%, respectively). Streptococcaceae was recorded with high relative abundance only in slaughterhouse female (7.51%) and male (5.37%) ticks. The bacterial genus, *Proteus* was detected with the highest relative abundance in male ticks collected from farms (89.83%); however, *Francisella* showed the highest relative abundance in female ticks collected from slaughterhouses (70.55%). In male ticks collected from slaughterhouses, *Francisella* was identified with high relative abundance (29.56%) followed by *Staphylococcus* (21.89%). The bacterial genus *Corynebacterium* was found with high relative abundance (31.74%) in female ticks collected from farms followed by *Moraxella* (21.97%). The *Fusobacterium* had low relative abundance 0.02–2.3%, and, also, *Escherichia* showed a low relative abundance of 0.01–0.39% in all ticks from all habitats except farm male ticks, where both were absent. *Amycolatopsis*, *Faecalibacterium*, and *Paraliobacillus* were recorded only in female farm ticks, with a low relative abundance of 0.10–0.18%.

### 3.3. Microbial Community Diversity

We evaluated variations in bacterial diversity between tick samples collected from slaughterhouses and farms. An increase in the diversity of slaughterhouses compared with farms was observed for all the metrics on alpha diversity (Appendix A).

Principal Coordinates Analysis showed that coordinates 1 and 2 accounted for over 98% of the variation (based on cumulative Eigenvalues), and the first two coordinates accounted for over 97% of the variation. Furthermore, there was a separation among the microbial communities between camel tick habitats, farms and slaughterhouses, and also between farm female and farm male ticks (Figure 3). Hierarchical clustering of samples based on Jaccard distance showed that the microbiome of farm female and farm male clustered separately, while slaughterhouse females and slaughterhouse males clustered closely together (Appendix A).

### 3.4. Impact of Tick Habitat on Bacterial Community Assembly

We evaluated the impact of tick habitat on the bacterial community’s assembly using co-occurrence networks. Overall, the results reveal significant differences in microbial interactions and the structure of microbial communities in each habitat (Figure 4, Appendix A).

In the farm network, 113 nodes and 758 edges were identified, while the slaughterhouse network shows greater complexity, with 219 nodes and 2116 edges. This suggests that the slaughterhouse environment supports a higher diversity of microbial interactions. Regarding positive and negative associations, 41.95% of connections in the farm are positive, while this figure rises to 49.24% in the slaughterhouse, reflecting a more balanced environment where positive microbial associations might be more prevalent. Positive correlations often suggest cooperative or mutually beneficial interactions, such as nutrient exchange, co-metabolism, or habitat sharing, which may enhance microbial resilience or stability. In contrast, negative correlations can indicate competitive relationships, niche exclusion, or antagonistic effects, such as the production of antimicrobial compounds by one species that inhibit others. The higher prevalence of positive correlations in the slaughterhouse environment may result from reduced environmental stress or the availability of more stable resources. These conditions promote microbial coexistence and synergy. Additionally, the balanced microbial environment in the slaughterhouse could play a crucial role in suppressing potential pathogens through mechanisms like competitive exclusion or the enhancement of beneficial microbial consortia. The negative modularity in both environments (−2675 in the farm and −10,212 in the slaughterhouse) suggests that the microbial communities are not well-defined, implying that associations within the microbiota tend to be diffuse and spread across different subgroups. However, the slaughterhouse has fewer communities than the farm (44 compared to 51), indicating a higher cohesion among microorganisms in that environment. The network diameter, which measures the longest distance between nodes, is slightly larger in the slaughterhouse (three, compared to two in the farm), indicating that connections between different microorganisms may be more dispersed in this setting. This is reinforced by a higher average degree in the slaughterhouse (19.32, compared to 13.41 in the farm), implying that nodes in the slaughterhouse tend to interact with more taxa. Additionally, the clustering coefficient and the number of triangles are both higher in the slaughterhouse (0.67 and 29,716 compared to 0.43 and 4646 in the farm), suggesting that interactions in the slaughterhouse are more complex and microorganisms tend to form denser groups, which could influence the stability of the microbial communities. The results suggest that microbial community associations in the slaughterhouse are denser and more interconnected, indicating a more favorable environment for the coexistence and cooperation among different microorganisms. In contrast, the farm network exhibits a less interconnected and more fragmented structure, which could reflect a microbial community more vulnerable to disturbances.

One notable difference was observed in the interactions involving *Francisella* (Figure 4). In the farm network, *Francisella* interacted with a diverse set of taxa, including Streptomyces, Alphaproteobacteria, and Bacteroides, all of which were located within the same microbial community. In contrast, in the slaughterhouse network, *Francisella* was found to interact only with uncultured Peptostreptococcaceae, which was located in a different community. Importantly, all interactions involving *Francisella* were positive in both environments. This shift in *Francisella*’s microbial partners between environments may reflect changes in the ecological roles or environmental pressures faced by *Francisella* in these two settings. In the farm, *Francisella* is part of a broader, more interconnected community with a variety of symbiotic relationships. However, in the slaughterhouse, its interactions are more limited, suggesting that specific environmental conditions or microbial dynamics are favoring a more specialized interaction. The fact that these interactions remain positive in both settings highlights *Francisella*’s potential role as a stable, cooperative member of the microbiota, contributing to the fitness of its microbial partners in different ecological contexts.

We also performed a comparison of microbiome studies in different *Hyalomma* tick species in the MENA region (Appendix A). A total of seven studies were conducted on *H. dromedarii* and *H. anatolicum*, and *Francisella* was found with high abundance in most of the studies.

## 4. Discussion

This study provides the first comprehensive assessment of microbial diversity in male and female camel ticks and estimates of microbial patterns across different habitats including farm and slaughterhouses. Previous studies focused only on female camel ticks collected from different farms [37,46]. Tick microbiota play an important role in tick nutrition, development, reproduction, and pathogen transmission and vector competence [4,41,67]. Therefore, it was crucial to evaluate the *H. dromedarii* microbiome for devising mitigation strategies, as this tick species is the most prevalent in the UAE.

The initial establishment of the microbiome in ticks occurs through transovarial transmission, wherein the adult female tick transfers microbes to her offspring. Beyond this, ticks can acquire microbes from their environment and through blood feeding on vertebrate hosts. Bacteria can enter the tick primarily via transovarial, oral, or cuticular routes [68]. The tick microbiome may be influenced by physiological adaptations that support prolonged blood feeding. Additionally, bioactive molecules in tick saliva modulate the host’s immune and inflammatory responses, facilitating pathogen acquisition [69]. Our findings highlight that ticks collected from the slaughterhouse have more diverse microbial communities as compared to farm-collected ticks, which may be attributed to the differences in environmental stressors (which could be temperature and relative humidity fluctuation, animal blood and wastes, feces of animals or animal manure, etc.) and host exposure. In terms of sex-based microbial diversity, male ticks from slaughterhouses displayed higher diversity than both male and female ticks from farms. It is well-established that geographical location and environmental factors influence the type of microbiota in ticks [15,70,71]. As a result, we can expect microbes to adapt to their specific environmental conditions, potentially leading to variations in microbiota composition. Microclimate and host factors played an important role for a subset of the tick microbiome [72].

The relative abundance of bacterial phyla in the current study differs from previous findings. Pseudomonadota was the most abundant phylum, followed by Bacillota and Actinomycetota, contrasting with earlier reports where *Proteobacteria* was the most abundant followed by *Firmicutes* in *H. dromedarii* ticks [37,44,46,73]. In addition, Pseudomonadota, Bacillota, and Actinomycetota were found to be abundant in all habitats and both sexes in the present study. This discrepancy might be explained by geographical differences, environmental factors such as temperature, and the type of sampling sites used [34]. For example, the bacterial microbiome composition of *Ixodes scapularis* was found to have the highest relative abundance of Proteobacteria under different temperatures in both male and female ticks [74]. In another study, Proteobacteria, Firmicutes, and Actinobacteriota were found to be dominant in the microbiome of *Rhipicephalus linnaei* and *Haemaphysalis leachi* [75]. The dominance of Proteobacteria across multiple studies suggests that habitat and climate greatly impact microbial composition. These results contradict with the results of *I. scapularis,* which may be due to environmental factors, seasonality, and habitat impacting on the patterns of tick-borne microbes.

At the class level, we identified 17 bacterial classes, with Gammaproteobacteria, Actinomycetes, Bacilli, and Tissierellia being the most abundant. These results align partially with studies conducted on *H. dromedarii* in the UAE and *Haemaphysalis* ticks in Malaysia [76], where Gammaproteobacteria, Bacilli, and Actinobacteria were also dominant. The variation in the dominance of different bacterial families, such as Morganellaceae in farm male ticks and Francisellaceae in slaughterhouse ticks, further underscores the influence of habitat and sex on microbiome composition. Interestingly, the high prevalence of Morganellaceae in male farm ticks deviates from earlier findings where Francisellaceae was dominant in female farm ticks [46]. This suggests that male ticks may have distinct ecological or behavioral traits that influence microbial colonization, such as differences in blood-feeding patterns or environmental exposure.

Regarding bacterial genera, the genus *Proteus* showed the highest relative abundance in male ticks collected from farms, whereas *Corynebacterium* was found with high relative abundance in female ticks collected from farms. Our results are consistent with previous findings on temporal changes in microbial communities of female *H. dromedarii* ticks where the genus *Corynebacterium* was found with a high relative abundance throughout the year [37]. In slaughterhouse-collected ticks, *Francisella* was found with highest relative abundance in both female and male ticks. These findings highlight the differences in the bacterial diversity of *H. dromedarii* in the UAE and supports the understanding that tick sex and habitats/environmental surroundings affect microbiome composition [35,77]. The genus *Francisella* has been previously detected in *H. dromedarii* ticks from Palestine, Saudi Arabia, and the UAE [44,45,46]. The results also revealed that several bacterial genera coexist in *H. dromedarii,* suggesting that they flourish under similar conditions and microbial interactions inside the tick host, resulting in the dominance of some genera over others. The bacterial genus *Proteus* showed a significant negative association with other genera.

The significant negative associations between *Proteus* and other genera, as well as the interactions of *Francisella* with other microbial taxa, suggest potential competitive or inhibitory relationships within the tick microbiome. These microbial interactions could have important implications for pathogen transmission and vector competence [36,78,79]. For example, the negative interaction between *Proteus* and *Uruburuella* may reduce the tick’s burden of pathogenic bacteria, thereby influencing the likelihood of disease transmission to the host. Conversely, symbiotic bacteria like *Francisella* could play a protective role by outcompeting harmful pathogens, potentially enhancing the tick’s survival and reproductive success [80,81]. The positive association between *Francisella* and *Streptococcus* suggests that some genera may coexist or even cooperate, supporting the tick’s biological functions. Understanding these dynamics is crucial for developing more effective tick control strategies. By targeting key microbial interactions—such as enhancing the presence of symbiotic bacteria like *Francisella* or manipulating competitive relationships involving *Proteus*—we could reduce the tick’s capacity to transmit harmful pathogens and improve vector control efforts. However, this kind of pathogen management strategy involves multiple laboratory experiments.

Our study reveals that samples collected from slaughterhouses had higher bacterial diversity compared to tick samples collected from farms, supporting the statement that habitat plays a major role in shaping tick microbiota. A previous study indicated ticks may acquire bacteria from habitats and blood meals [82]. In slaughterhouse-collected ticks, the microbiomes of males were more diverse than those of females, while it was opposite in farm-collected ticks, suggesting complex sex-specific and habitat-specific influences on microbial communities. Tick-associated bacterial communities of male ticks appeared to be more diverse than those of adult females, potentially due to differences in behavior and physiology [77,83]. Tick species microbiomes vary with sex and habitat ranges, suggesting that exposure to environmental conditions/stress, tick immunity, host, and blood meals may influence the tick microbiome, thus affecting pathogen transmission [34,84,85,86,87].

Microbiome studies are enhancing our understanding of the relationship between tick microbiome structure, endosymbiont interactions [88], and tick vector competency [84]. While NGS technologies have significantly deepened our insights into microbial community dynamics and ecology, the focus on higher taxonomic levels (such as bacterial genera) can sometimes limit the detection of pathogens and obscure the identification of pathogen–microbiota interactions, as species-level identification remains challenging [89]. In the MENA region, seven microbiome studies have been conducted on *H. dromedarii* and *H. anatolicum* tick species [37,44,45,46,71,73,90], underscoring *Francisella*’s potential role as a stable and collaborative member of the microbiota, contributing to the fitness of microbial partners in various ecological contexts. NGS studies provide valuable baseline information on the microbial communities associated with ticks across different ecosystems and highlight the need to screen detected bacterial genera for potential pathogens and endosymbionts.

## 5. Conclusions

In conclusion, the bacterial microbiomes of *H. dromedarii* were distinct between male and female ticks in different habitats. Previous studies focused on the farm female camel tick microbiome, and our results about the farm female tick microbiome are consistent with previous findings. However, in this study, we found *Proteus* with a high relative abundance in farm male ticks. *Corynebacterium*, *Staphylococcus*, *Peptoniphilus*, and *Moraxella* were abundant bacterial genera in both habitats. In addition, slaughterhouse-collected ticks had more diverse microbial communities as compared to farm-collected ticks, though microbial community associations in the slaughterhouse were denser and more connected as compared to farms. Recent findings may provide insight into why ticks in different habitats vary in their ability to acquire, carry, and spread pathogens. This is preliminary work for future large-scale comparative tick microbiome studies with regards to tick stages, sex, and environmental settings.

## Figures and Tables

**Figure 1 insects-16-00011-f001:**
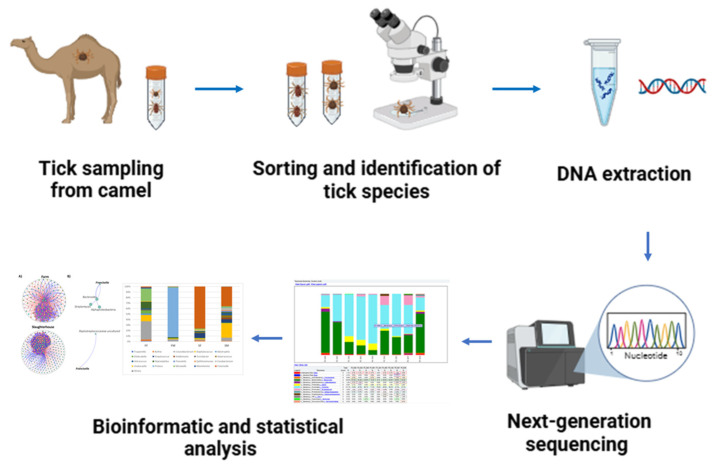
The layout of working procedure for data collection and analysis. The figure was created with BioRender (https://Biorender.Com/, accessed on 16 November 2024).

**Figure 2 insects-16-00011-f002:**
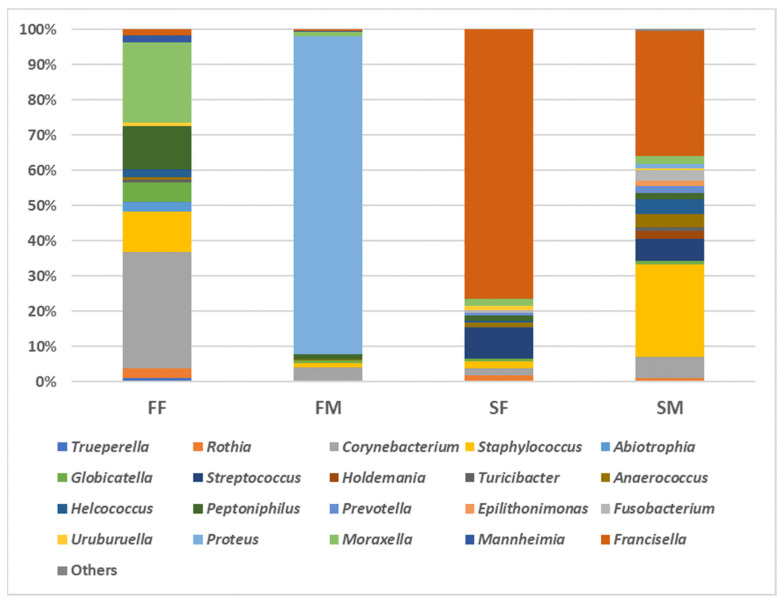
Microbial genera detected in *H. dromedarii* ticks collected from different habitats, farm and slaughterhouse, respectively. Abbreviations: FF, Farm Female; FM, Farm Male; SF, Slaughterhouse Female; SM, Slaughterhouse Male.

**Figure 3 insects-16-00011-f003:**
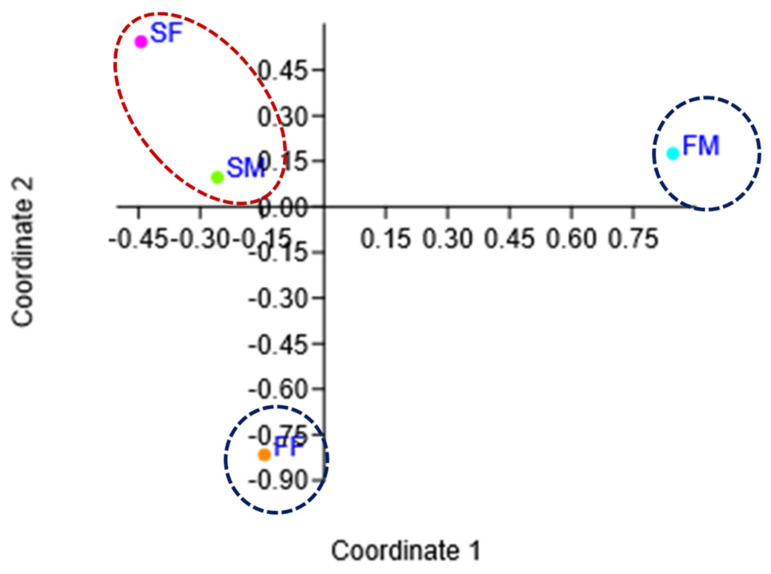
Principal Coordinates Analysis showing microbial diversity in ticks collected from different habitats. Blue and red circles refer to the habitat, farm and slaughterhouse, respectively. Abbreviations: FF, Farm Female; FM, Farm Male; SF, Slaughterhouse Female; SM, Slaughterhouse Male.

**Figure 4 insects-16-00011-f004:**
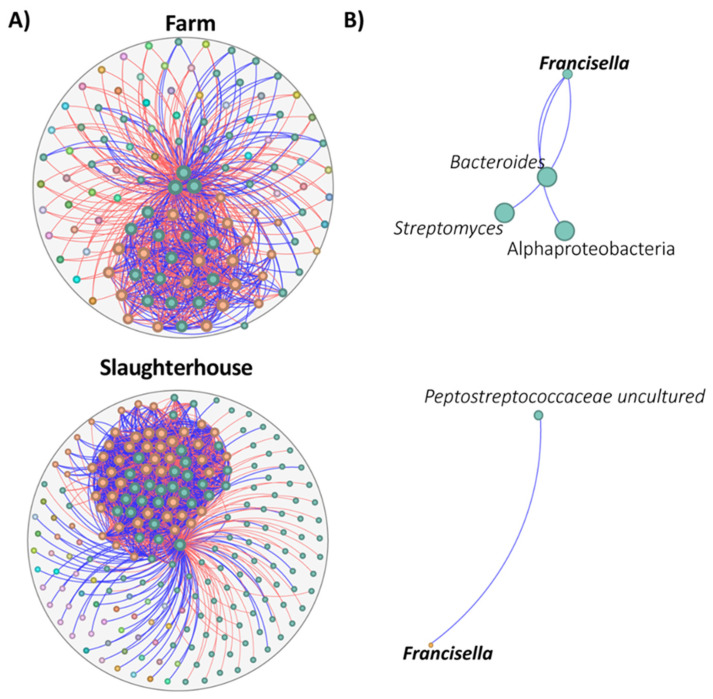
Taxonomic networks representing different on-the-tick habitats. (**A**) Bacterial co-occurrence networks of farm and slaughterhouse. (**B**) Sub-networks of the local connectivity of *Francisella* in farm and slaughterhouse networks. Node colors are based on a modularity class metric; each module is represented by a different color. The size of nodes is related to their eigenvector centrality; the bigger the node, the higher eigenvector centrality value it has. Positive (blue) or negative (red) correlations are shown by the color of the edges. Bacterial taxa (family or genus level) with at least one connection are symbolized by nodes, whilst connected edges represent correlations between them (SparCC ≥ 0.75 or ≤−0.75).

**Table 1 insects-16-00011-t001:** Detail of DNA pools.

List of Pools	Pools	Sample ID
Farms/females	1st pool	FF
Farms/males	2nd pool	FM
Farms/nymphs	3rd pool	FN
Slaughterhouse/females	4th pool	SF
Slaughterhouse/males	5th pool	SM
Slaughterhouse/nymphs	6th pool	SN
Livestock market/females	7th pool	LF
Livestock market/males	8th pool	LM

## Data Availability

All data are contained within the manuscript and Appendix A.

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
