# Peer review of "Microbiome of Hyalomma dromedarii (Ixodida: Ixodidae) Ticks: Variation in Community Structure with Regard to Sex and Host Habitat"

_insects, 2024, doi:10.3390/insects16010011_

Round 1
Reviewer 1 Report
Comments and Suggestions for Authors
1. The principal question is – why did You research ticks collected from animals, not from vegetation? Ticks intended to analyse were partially engorged. How can You claim that the determined microbiome is specific to Hyalomma dromedaria? Variation in bacterial community structure may result from what was present in the host's blood.
2. Tick sampling Materials and methods
It was written that ticks were collected in 2022 – but which month?
According to article by Perveen et al. (2022; Front Vet Sci), the microbiome changes in different months in H. dromedarii partially engorged female ticks. Moreover, the discussion does not contain any references to the results of this previous research - i.e., to the time at which ticks were obtained for research.
3. Discussion
L 382-383: there is no citation (I think it should be: Perveen et al. (2022; Front Vet Sci)
What new does this work bring due to the fact that additional males were examined, unlike the previous work where only females were examined?
L: 389: what kind of environmental stressor?
General comment: In discussion there are many repetitions of the same theses - especially that the microbiome influences the transmission of pathogens - but without providing an example (L88-89, L384, L 436-437, L459, L471).
There is often reference to climate change as a factor influencing bacterial diversity - but how exactly? These types of statements are too laconic and do not bring anything new to the work.
L 470-471: I disagree with this statement. This was not the purpose of this study, and the results do not support this thesis.
Conclusion part – This is the weakest part of the article, there is no clear indication of what the result of this work is, there is no clearly defined conclusion - the same applies to the abstract.
Reviewer 2 Report
Comments and Suggestions for Authors
This work makes a significant contribution to our knowledge of the tick microbiome, which remains poorly understood. The study is substantial and well-prepared. However, it requires numerous editorial corrections, and I also have several substantive comments.
I will begin with the editorial remarks:
- spaces: Line 79 - double space after T. camelensis and an extra space after Sindbis; lines 137 and 148 - there should be no space before °C; Line 232 - H. drome- - extra space; line 254 - extra space before "(Supplementary Figure 1; Table 4)"; lines 291 and 292 - missing spaces near numerical values; line 303 - extra space; line 304 - missing space in "2.1in"; line 330 - extra space; line 384 - extra space
- The description of Figure 1 is missing "SM."
- In lines 405, 407, and 414, citations use author names instead of numerical references.
- Line 409 - it should be I. scapularis. There is an unnecessary period, a missing space, and then a double space.
Now, a few smaller or more significant substantive comments:
- Line 131 - "Ticks were placed in Eppendorf tubes (50mL) and brought to the Parasitology Laboratory" - Are you sure you used 50 mL tubes and not 1.5 mL?
- Abstract - In my opinion, it needs revision: in the abstract, you state that you collected a total of 40 males and females, but in the text, it says, "DNA was extracted individually from 40 ticks (male (15), female (15) and nymphs (10))." In the abstract, you also mention the year 2023, but in the text, there is no reference to it. Only the year 2022 is mentioned.
- It's a pity that you assumed in advance that ticks collected from this environment would have the same microbiome and pooled them together. I’m curious whether this is actually the case, or if the microbiome might differ within a single environment.
- If only 4 pools passed the quality control, why weren’t they repeated? Why did you collect as many as 350 ticks but isolate DNA from only 40? If they weren’t used for the study, why mention them? On the other hand, if you collected so many, you had sufficient material to repeat the sequencing. Information about the microbiome in nymphs would have greatly complemented the considerable effort you put into this article.
- Lines 444-447 - "By targeting key microbial interactions—such as enhancing the presence of symbiotic bacteria like Francisella or manipulating competitive relationships involving Proteus—we could reduce the tick's capacity to transmit harmful pathogens and improve vector control efforts." - Do you really intend to control vectors by increasing the presence of Francisella in ticks? While Francisella may indeed be symbiotic for ticks, Francisella tularensis is a pathogen, and the suggestion to increase the presence of pathogens to reduce the presence of pathogens seems controversial to me.
The work is good. With minor corrections, it could be very good.
Reviewer 3 Report
Comments and Suggestions for Authors
Perveen et al. present an interesting study on the microbiome of Hyalomma dromedarii (Acari: Ixodida: Ixodidae) ticks, including variation due to sex and habitat. The study had a particularly interesting background in the context of camel husbandry in the UAE. The attempt was commendable to capture microbiota diversity using NGS in ticks collected from farms and slaughterhouses. However, a number of aspects need revisions for further clarity, rigor, and consistency. The details of the assessment and critical points/suggestions to improve the scientific robustness and readability for this work are hereby presented.
Major comments:
1. Explanation of the sex-based microbiome findings: Although the manuscript indicates that Proteus dominates in the male ticks collected from farms while Corynebacterium and Francisella dominate in other environments or sexes. This should be further elaborated to explain better the observed sex- and habitat-based differences in bacterial dominance. Probably the underlying factors or physiological aspects that can drive these differences may be specified. For example, feeding behavior or immunity.
2. Statistical validation: Comparison of microbial diversity indices, richness, and evenness should be appropriately mentioned to be valid in conditions when microbiome data are not normally distributed. Normality check and, whenever applicable, non-parametric alternatives are encouraged as a means of strengthening statistical assertions of richness and evenness.
3. Genus-level interaction accuracy: Network analysis and descriptions of correlations denote complex interactions within the microbiome, especially with Francisella. This report uses Pearson correlations, but further description is warranted regarding whether co-occurrence methods like SparCC may be more informative regarding microbial relationships. For clarity in methodology, traditional Pearson correlation will poorly capture compositional relationships in microbiome data.
4. Habitat influence interpretation: The authors conclude that slaughterhouses are a niche with higher microbial diversity due to the environmental stressors, while it should refer more precisely to what specific environmental variables (such as hygiene practices or animal density), or host factors may explain the bacterial diversity. A clear causal linkage will help to support the conclusion better.
Moderate points to consider:
1. Methodological transparency: Pooling of the tick samples for DNA extraction needs to be supported since the combined microbiota represents multiple independent ecologies. Such pooling may mask variability in individual microbiomes. The justification of pooling has to be done by providing very appropriate references, or the proposal of follow-up studies for analyzing individual ticks could be presented.
2. OTU-based limitations: This study has employed OTUs instead of ASVs. So, limitations in taxonomic resolution might affect pathogen detection. Discuss this methodological choice and its possible implications for identifying rare pathogenic species.
3. Ecological relevance of positive and negative correlations: There is a need to explain what positive and negative bacterial correlations mean, as most of them are driven by the physiological ambiance of the tick. It would have been great if ecological relevance or mechanisms of potential pathogen suppression could be added in order to help readers understand such dynamics more clearly.
Minor inquiries:
Data deposition details: Although the manuscript was supported with a BioProject ID with regards to data deposition, metrics for raw data quality control are not available. It would bring more transparency if information such as sequencing depth and statistics about the quality of the reads were reported.
Consistency in taxonomic presentation: The presentation of taxonomy for some bacteria, such as Proteus and Francisella, is not consistent, sometimes using genus and sometimes family. This should be harmonized throughout for readability and to avoid ambiguity.
Inclusion of supplemental material: There is some very useful quantitative information about microbial abundance in the supplemental tables. Perhaps briefly summarize some of the main conclusions from supplemental tables in the text to make it easier to follow w/o switching back and forth.
Thus, this interesting study provides for the first time important data on the microbiome of the ixodid H. dromedarii and reveals differences influenced by sex and habitat. Consequently, with these essential revisions to enhance clarity in methodology and interpretation of results, the manuscript will meet the standard of publication and constitute a worthier contribution to research into the microbiomes of ticks.
Text edits:
Line 20: Change precious to highly valued.
Line 21: Change causing to leading to.
Line 23: Change to understand how to to investigate how.
Line 25: Change is essential to prevent to is essential for preventing.
Line 33: Change were taken from tick samples collected from camels to were selected from samples collected from camels.
Line 121-122: Change We obtained permissions for tick collection within the studied areas to Permission for tick collection was obtained from relevant authorities.
Line 136: Change Ticks samples were kept to Tick samples were stored.
Line 167: Change assign_taxony.py to assign_taxonomy.py.
Line 182: Change Bray–Curtis evaluated the dissimilarity to The Bray–Curtis index was used to evaluate dissimilarity.
Line 190: Change Networks provide to The networks provide.
Line 195: Change The colors of nodes were assigned to Node colors were assigned.
Line 359: Change are denser and more connected to are denser and more interconnected.
Line 107: Change harbor to harbors.
Line 153: Change community pattern in camel ticks to community composition in camel ticks.
Line 255: Change The Francisellaceae was present with high relative abundance to Francisellaceae was highly abundant.
Line 380: Change represents the first comprehensive assessment to provides the first comprehensive assessment.
Line 387: Change collected from the slaughterhouse had more diverse to collected from the slaughterhouse have more diverse.
Line 391: Change It is well established that to It is well-established that.
Line 428: Change was detected previously to has been previously detected.
Reviewer 4 Report
Comments and Suggestions for Authors
Overall, this manuscript appears to be simplistic, primarily focusing on the variations in the microbiome of Hyalomma dromedarii ticks between different sexes and host habitats. While it presents the composition of microbial communities detected in ticks collected from various environments and discusses their correlation with pathogens and endosymbionts, the nature of these correlations remains unclear. Please consider removing some results that are not needed for the description. I have the following concerns:
1. The original microbiome data is publicly available. Please also layout the working procedure for the data collected/analysis, which can be illustrated in the main figure.
2. In addition to the correlation analysis presented in Fig. 3, I recommend including additional analyses, such as data robustness assessments, comparisons with real experimental data (which is already online with other people’s data, such as other related species, their ticks microbiome data for comparing), or microbial sampling results.
3. The figures and results presented in the manuscript are relatively simplistic. I suggest avoiding the use of bright red or blue colors, as they detract from the overall aesthetic quality.
4. Additionally, when the results are limited and there are no hypotheses or speculations presented, it becomes hard to form a cohesive idea throughout the MS.
Comments on the Quality of English Languagenone
Author Response
Please see attachement.

Round 2
Reviewer 1 Report
Comments and Suggestions for Authors
I accept the authors' explanations, corrections have been made in the body text.
Author Response
Comments: I accept the authors' explanations, corrections have been made in the body text.
We thank the reviewer for accepting our explanations and corrections in the body text.
Reviewer 4 Report
Comments and Suggestions for Authors
Overall, substantial revisions were made to the MS. However, I noticed that the quality of the figures remains not good. I strongly recommend that the authors provide vector-based PDF figures. Additionally, I find that the authors did not directly address the concerns I raised in Comment 2 and Comment 4.
Author Response
Dear Reviewer,
Thank you very much for taking the time to review this manuscript. Please find detailed responses below and corresponding revisions/corrections highlighted in the re-submitted files to the additional comments and suggestions:
“Overall, substantial revisions were made to the MS. However, I noticed that the quality of the figures remains not good. I strongly recommend that the authors provide vector-based PDF figures. Additionally, I find that the authors did not directly address the concerns I raised in Comment 2 and Comment 4.”
Figures. Thank you for comment on the figures. We produced high quality 600dpi figures as per guidelines for authors on MDPI Insects’ website.
Comment 2. In addition to the correlation analysis presented in Fig. 3, I recommend including additional analyses, such as data robustness assessments, comparisons with real experimental data (which is already online with other people’s data, such as other related species, their ticks’ microbiome data for comparing), or microbial sampling results.
Response 2: We removed correlation from the manuscript. We appreciate your suggestion; however, this data analysis will produce a review paper on comparison of Hyalomma ticks’ microbiome which we will complete in our next project. However, we have compared the data and produced a table to compare microbiomes of Hyalomma ticks and added in supplementary data table 7. This is the last month of our project. Therefore, it will be challenging for us to extend our work. We hope you will consider our request.
Comment 4: Additionally, when the results are limited and there are no hypotheses or speculations presented, it becomes hard to form a cohesive idea throughout the MS.
Response 4: We have added a hypothesis to make reading smooth (116-117). We agree with you, results are limited because our four samples could not pass quality control test (samples from livestock market and life stage nymph). However, we plan to conduct a large-scale study from all over the UAE on camel tick microbiome, how pathogens and symbionts interact within camel tick species. Furthermore, we also plan to study microbial community composition in different stages including eggs, larvae, and nymphs to make assessments of transovarial and transstadial transmission of pathogens and endosymbionts.
Round 3
Reviewer 4 Report
Comments and Suggestions for Authors
The authors have almost addressed my concerns.